# Differential Phosphorus Allocation in Leaves and Roots of *Yushania shuichengensis* Across Soil Phosphorus Gradients: Implications for Ecological Adaptation and Resource Use Efficiency

**DOI:** 10.3390/biology14121647

**Published:** 2025-11-23

**Authors:** Shun Zou, Chumin Huang, Xiaolong Bai, Wangjun Li, Bin He

**Affiliations:** School of Ecological Engineering, Guizhou University of Engineering Science, Bijie 551700, China; zoushun@gues.edu.cn (S.Z.); huangcm@gues.edu.cn (C.H.); baixiaolong@gues.edu.cn (X.B.); teesn470@gues.edu.cn (W.L.)

**Keywords:** phosphorus fraction, phosphorus resorption efficiency, photosynthetic phosphorus use efficiency, root morphology, soil phosphorus gradient

## Abstract

Plants need P to grow, yet its scarcity in soils often limits ecosystem productivity. While plants regulate internal forms of P to cope with scarcity, how wild species utilize these forms under natural soil gradients remains unclear. This study investigated P distribution in leaves and roots of a wild bamboo (*Y. shuichengensis*) across low-, medium-, and high-P soils. We found that in leaves, P for energy and structure peaked at medium soil levels, while high P promoted storage of inactive forms. Root P increased directly with soil content. P reuse efficiency from aging leaves decreased with higher leaf P but was unaffected by internal redistribution. Photosynthesis relied on total P rather than internal reallocation. In roots, more P allocated to genetic material supported longer, more efficient roots. This bamboo adapts to soil P variation mainly by adjusting concentrations of key P forms in each organ, rather than redistributing them internally. These insights help guide sustainable management of alpine bamboo ecosystems by maintaining optimal soil P, reducing fertilizer use and environmental impact.

## 1. Introduction

Phosphorus (P) is an essential element that frequently limits plant growth, photosynthesis, and ecosystem productivity across diverse terrestrial ecosystems [1,2]. The availability of P is crucial for plant physiological processes, particularly in low-P environments such as highly weathered tropical soils and those developed on sandstone and limestone, where P deficiency can significantly constrain plant development [3,4]. To adapt to P limitation, plants optimize P-use efficiency through key mechanisms including P absorption, internal allocation, and resorption [5]. Among these, the allocation of P to different biochemical fractions—inorganic P (P_i_), metabolite P (P_M_), lipid P (P_L_), nucleic acid P (P_N_), and residual P (P_R_)—represents a critical strategy for meeting metabolic, structural, and storage demands under varying soil P conditions.

Extensive research has sought to decipher how plants adjust P allocation in response to soil P availability. A common paradigm suggests that under low P supply, plants increase the proportion of P allocated to metabolic pools (P_M_, P_L_) to enhance photosynthetic capacity, whereas under high P conditions, they accumulate Pi as a storage pool [6,7]. However, a growing body of evidence from field studies reveals considerable complexity and context-dependency in these patterns, challenging the generality of this paradigm. For instance, along natural soil P gradients, Australian plants on low-P sandstone soils showed a higher proportion of P_L_, while in Malaysian montane forests, no significant differences in leaf P fraction proportions were found across varying soil P concentrations [5,7,8]. Similarly, P-addition experiments have yielded inconsistent results: some report a reduction in P_N_ and P_R_ allocation [9], while others in subtropical forests show an increase in P_M_ proportion alongside a decrease in other fractions [10]. These conflicting findings underscore a significant knowledge gap: we lack a unified understanding of how P allocation strategies are shaped by intrinsic (e.g., species-specific) and extrinsic (e.g., environmental gradient) factors in natural settings.

The relationship between P allocation and physiological efficiency further illustrates this complexity. Studies on crop genotypes with high P-use efficiency consistently show that plants maintain photosynthetic rates under low-P conditions by reallocating P fractions (e.g., reducing investment in P_i_ and P_L_), thereby enhancing photosynthetic P-use efficiency (PPUE) [11,12,13]. However, this consistent pattern is seldom observed in wild plants. In subtropical trees, for example, the maximum net photosynthetic rate (Pn) was positively correlated with all P fractions, and PPUE was positively correlated with structural P (P_R_) but negatively with metabolic P [14]. A recent study even reported a negative correlation between Pn and both total P and PL [15], contradicting the expectation that higher P supply supports higher photosynthesis. This stark contrast between controlled agricultural systems and natural ecosystems highlights the pressing need to investigate P allocation ecology in non-domesticated species, particularly under natural environmental gradients.

Despite the ecological importance of bamboo, their P allocation strategies remain almost entirely unexplored. This study addresses this critical gap by investigating *Y. shuichengensis* Yi and L, a typical alpine bamboo species, across a natural soil P gradient in Southwest China [16]. The study area, characterized by diverse parent rocks (sandstone, limestone, basalt) under similar climatic conditions, provides an ideal natural experiment to examine plant responses to varying soil P availability [17]. Given the conflicting evidence and the lack of data for bamboo, we reframe our investigation around the following questions: (i) How do soil P availability levels influence the concentrations and allocation patterns of different P fractions in the leaves and roots of *Y. shuichengensis*? While resource optimization theory might predict increased allocation to metabolic fractions under low P, the specific patterns in this bamboo species remain an open question. (ii) Do leaves and roots exhibit distinct P allocation priorities reflective of their organ-specific functions? (iii) To what extent are key functional traits (Pn, PPUE, P resorption efficiency, root morphology) related to the internal P allocation strategies? Specifically, is P fraction reallocation a key lever for maintaining efficiency, or do other mechanisms dominate? By addressing these questions, this study aims to provide novel insights into the P allocation ecology of bamboo, clarifying how wild plants fine-tune P-use strategies to adapt to soil P variability.

## 2. Materials and Methods

### 2.1. Study Site

The study area is located in the Jiucaiping Mountain Range on the northeastern Yunnan–Guizhou Plateau in Guizhou Province (104°39′15″–104°44′46″ E, 26°50′39″–27°0′28″ N). All plots are at an elevation of ≥2200 m a.s.l., with climate dominated by topographic elevation rather than latitude. Although situated within the subtropical monsoon zone, the mountain summit features a cool and humid subalpine climate. Data from the 2400 m automatic weather station (1991–2020, Bijie Meteorological Bureau) show a mean annual temperature of 8.1 °C (−1.7 °C in January and 16.9 °C in July) and an annual precipitation of 1094 mm, with more than 75% concentrated from May to September. The dominant vegetation on the ridges and saddles is the bamboo shrub *Y. shuichengensis* and subalpine meadows.

Bedrock alternates over distances of kilometers, including Permian Emeishan Large Igneous Province basalts, Maokou Formation limestones, and contemporaneous sandstones [17,18,19]. Subalpine meadow soils developed from different parent materials show significant differences in pH, base saturation, and nutrient content, with the total P gradient being the most pronounced: 2.73 g kg^−1^ in basalt derived soils, 1.60 g kg^−1^ in limestone derived soils, and only 0.66 g kg^−1^ in sandstone derived soils (Table 1).

The three study sites were selected to represent the low, medium, and high ends of the natural soil total P (TP) gradient within the primary distribution area of *Y. shuichengensis* on the Yunnan-Guizhou Plateau, based on a preliminary regional soil survey. These sites were chosen for their highly comparable climatic conditions and minimal historical anthropogenic disturbance, ensuring that the soil P gradient was the primary variable influencing plant physiology. The measured TP values (0.66, 1.60, and 2.73 g kg^−1^; Table 1) effectively span the dominant range of soil P availability in these ecosystems.

### 2.2. Field Sampling

Across the study region we first mapped monodominant *Y. shuichengensis* thickets anchored exclusively on sandstone, limestone or basalt by rapid floristic and pedological surveys. Five 5 × 5 m plots were positioned on each parent material (*n* = 15), separated by ≥200 m to ensure geostatistical independence. During peak growing season (July 2024) we performed in situ leaf gas-exchange measurements and collected plant material.

At each plot, topsoil (0–20 cm) was sampled with a stainless-steel auger using a five-point bulked composite per plot. Freshly senesced leaves were trapped with a 1 × 1 m litter frame, sealed in polyethylene bags and transported to the laboratory. Five outwardly intact ramets were then selected inside the plot. After completing non-destructive gas-exchange recordings (see below), we excised 40 fully expanded leaves and an entire root system from every individual. Each of the resulting 30 samples (3 lithologies × 2 organs × 5 replicates) was split: one subsample was kept moist and at a low temperature inside an insulated box with ice packs for functional-trait assays, while the other was rinsed with ultrapure water, snap-frozen in liquid nitrogen within minutes of harvest, and stored at –80 °C until P fraction determination.

### 2.3. Measurement of Functional Traits of Leaves and Roots

#### 2.3.1. Leaf Traits

On clear mornings in July 2024 (08:30–11:30), leaf gas exchange was measured with a Yaxin-1105 portable photosynthesis-fluorometer (Beijing Yaxinliyi Science and Technology Co., Ltd., Beijing, China). Leaves were equilibrated for 15 min at 1500 μmol photons m^−2^ s^−1^ (LED), 420 ppm CO_2_ and 25 °C before recording Pn (μmol m^−2^ s^−1^). Five intact, fully expanded leaves per plot were averaged. Projected leaf area was then scanned (HP Scanjet M231, HP Inc., Palo Alto, CA, USA) and analyzed (Image J 2, National Institutes of Health, Bethesda, MD, USA), and dry mass was obtained after 48 h at 70 °C (0.0001 g precision) to calculate specific leaf area (SLA, cm^2^ g^−1^) and convert Pn to a mass basis (nmol g^−1^ s^−1^) [20].

#### 2.3.2. Root Traits

Four key root architectural traits were quantified: specific root length (SRL, cm g^−1^), specific root surface area (SRA, cm^2^ g^−1^), mean root diameter (RD, mm) and root tissue density (RTD, g cm^−3^). From every sealed bag we retrieved an intact subsample, gently washed away adhering soil and spread the roots in a thin water film inside a transparent tray. High-resolution images were acquired with the same flatbed scanner driven by DJ-GX02 (2016C) imaging software (Dianjiang Tech, Shanghai, China). The program automatically exported total length, surface, volume and average RD. Material was then oven-dried at 70 °C for 72 h and weighed to 0.1 mg. SRL, SRA and RTD were computed as length, surface and volume divided by dry mass, respectively [20].

### 2.4. Measurement of P Fractions, and Total P

Soil samples were air-dried and passed through a 2 mm sieve. Soil TP was determined by iCAP 7400 ICP-AES (Thermo Fisher, Bremen, Germany) following perchloric-sulfuric acid digestion. Green leaf, senesced leaf and root material were oven-dried, pulverized to a fine powder and passed through a 60-mesh screen. Plant material TP was determined after acid digestion with an iCAP 7400 ICP-AES [21].

The sequential extraction of phosphorus fractions in green leaves and roots followed the protocol of Tsujii et al. [15] with minor adjustments. To ensure data accuracy, we accepted data only when the sum of the concentrations of all extracted P fractions fell within 90–110% of the TP content of the sample. Briefly, fresh materials were stored at −80 °C, lyophilized for 7 days, and ground to a fine powder. All subsequent steps were performed on ice using pre-chilled reagents. For the main fractionation, a 50 mg aliquot of powder was sequentially extracted: (1) three times with 3 mL of a chloroform–methanol–formic acid (CMF) solution to remove lipids; (2) the residue was re-extracted with 3.78 mL of a chloroform–methanol–water (CMW) mixture. The combined supernatants were mixed with 1.9 mL of chloroform–water mixture (CWW) for phase separation. The lower organic phase was collected as the phospholipid P (PL) fraction. The aqueous phase was saved. The pellet was then sequentially washed with 1 mL of 85% methanol and 2 mL of ice-cold 5% trichloroacetic acid (TCA, incubated at 4 °C for 1 h). The resulting supernatants were combined with the saved aqueous phase, constituting the primary metabolic P (PP_M_) pool, which contains inorganic P (P_i_) and other low-molecular-weight metabolic P (P_M_). The remaining tissue was hydrolyzed three times with 3 mL of 2.5% TCA at 95 °C for 1 h to extract nucleic acid P (P_N_). The final insoluble residue was defined as residual P (P_R_). All four extracts (P_L_, PP_M_, P_N_, P_R_) were dried under a stream of nitrogen, digested with 2.5 mL of concentrated H_2_SO_4_ and 1 mL of 30% H_2_O_2_ at 360 °C, diluted to 10 mL, and their total P content was determined by the molybdenum-blue method at 620 nm. For the specific quantification of Pi, 25 mg of powder was extracted three times with 0.5 mL of 1% glacial acetic acid on an ice bath, with centrifugation at 21,000× *g*. The combined supernatant was diluted six-fold, decolorized with acid-washed activated carbon, filtered through a 0.45-µm membrane, and analyzed by the malachite green method at 620 nm. The P_M_ concentration was calculated by subtracting the P_i_ concentration from the PP_M_ concentration. The relative allocation of each P fraction (rP_i_, rP_M_, rP_N_, rP_L_, rP_R_) was calculated as its proportion relative to the total P content in the respective organ.

### 2.5. Calculations of Photosynthetic P Efficiency and P Resorption Efficiency

PPUE (umol mol^−1^ s^−1^) was expressed as the net CO_2_ assimilation rate per unit leaf P mass (Pn/TP). Leaf P resorption efficiency (PRE, %) quantified the proportion of P remobilized before abscission and was calculated as the percentage difference between the P concentrations of green and senesced leaves relative to the P concentrations of green leaves [22].

### 2.6. Statistical Analysis

First, we used one-way ANOVA to test the differences in P fractions and their relative allocation ratios in leaves and roots across soil P gradients (low P, moderate P and high P sites). Levene’s test checked variance homogeneity, and data were log-transformed if needed. Duncan’s post hoc tests (*p* < 0.05) were applied. Second, paired *t*-tests were used to assess differences between leaves and roots in P fractions and their relative allocation ratios. Then, to explore the effects of total P concentration on P fraction concentrations and their relative allocation ratios, we fitted relationships using least-squares linear regression in both leaves and roots. Prior to regression analysis, the normality of residuals and homoscedasticity were checked using the Shapiro–Wilk test and visual inspection of residual plots, respectively. Finally, to determine the impact of P fractions and their relative allocation ratios on plant functional traits, we calculated Pearson correlations between these variables and functional traits in both leaves and roots. All statistical analyses were conducted in IBM SPSS Statistics 27 (IBM Corp., Armonk, NY, USA), and all figures were created in OriginPro 2024b (OriginLab Corporation, Northampton, MA, USA).

## 3. Results

### 3.1. Variation in P Fractions Among Sites

Leaf P fractions showed site independent concentrations for P_i_, P_N_ and P_R_ (Figure 1A,E,I), whereas P_M_ and P_L_ were equally elevated in moderate-P sites compared with low-P and high-P sites (Figure 1C,G). Relative allocations, by contrast, differed markedly across P gradient. High-P sites allocated the greatest proportion to P_i_ (rP_i_), whereas moderate-P sites favored P_M_ (rP_M_) and PL (rP_L_); rP_N_ and rP_R_ were again highest in high-P sites, with low-P sites being consistently intermediate (Figure 1B,D,F,H,J).

In roots all P fractions differed significantly across soil P concentration gradients. P_i_ ranked high-P site > moderate-P site > low-P site, while P_M_ peaked in moderate-P sites relative to low-P sites, and P_N_, P_L_ and P_R_ were equally higher in moderate-P and high-P sites than in low-P sites (Figure 1A,C,E,G,I). Allocation patterns were also site-dependent: high-P sites exhibited the greatest rP_i_, low-P and moderate-P sites showed higher rP_M_ and rP_L_, low-P sites yielded the highest rP_N_, and moderate-P sites displayed the largest rP_R_, with high-P sites intermediate for the latter (Figure 1B,D,F,H,J).

### 3.2. Variation in P Fractions Between Leaves and Roots

Paired *t*-tests revealed that foliar P_i_ (*p* < 0.001), P_N_ (*p* < 0.001), and P_L_ (*p* < 0.001) concentrations were significantly higher than those in roots, whereas P_M_ (*p* = 0.21) and P_R_ (*p* = 0.71) did not differ between organs. In terms of relative allocation, rP_N_ (*p* < 0.001) was greater in leaves, while roots exhibited higher rP_M_ (*p* < 0.001) and rP_R_ (*p* < 0.001); rP_i_ (*p* = 0.14) and rP_L_ (*p* < 0.43) showed no significant organ-dependent differences (Figure 1).

### 3.3. Correlations of TP with P Fractions in Leaves and Roots

Leaf TP scaled positively with P_i_ (slope = 0.27, *R*^2^ = 0.37, *p* = 0.01), P_M_ (slope = 0.40, *R*^2^ = 0.55, *p* < 0.001) and P_L_ (slope = 0.27, *R*^2^ = 0.49, *p* = 0.002), but showed no relationship with P_N_ (*R*^2^ = 0.09, *p* = 0.14) or P_R_ (*R*^2^ = 0.01, *p* = 0.32). Accordingly, TP was positively correlated with the proportional allocation to P_M_ (rP_M_, slope = 16.33, *R*^2^ = 0.40, *p* = 0.007) and negatively with rP_N_ (slope = −10.93, *R*^2^ = 0.35, *p* = 0.01); no significant trends were detected for rP_i_ (*R*^2^ = 0.02, *p* = 0.28), rP_L_ (*R*^2^ = 0.06, *p* = 0.19) or rP_R_ (*R*^2^ = 0.01, *p* = 0.32) (Figure 2).

In roots, TP exhibited a tight positive relationship with every P fraction (Figure 3): P_i_ (slope = 0.51, *R*^2^ = 0.83, *p* < 0.001), P_M_ (slope = 0.12, *R*^2^ = 0.24, *p* = 0.04), P_N_ (slope = 0.09, *R*^2^ = 0.78, *p* < 0.001), P_L_ (slope = 0.09, *R*^2^ = 0.50, *p* = 0.002) and P_R_ (slope = 0.11, *R*^2^ = 0.60, *p* < 0.001). Conversely, TP was positively associated with rP_i_ (slope = 18.66, *R*^2^ = 0.36, *p* = 0.01) but negatively with rP_M_ (slope = −12.15, *R*^2^ = 0.23, *p* = 0.04), rP_N_ (slope = −11.70, *R*^2^ = 0.76, *p* < 0.001) and rP_L_ (slope = −10.23, *R*^2^ = 0.53, *p* = 0.001); rP_R_ remained unrelated to TP (*R*^2^ = 0.04, *p* = 0.24).

### 3.4. Correlations of Soil TP with P Fractions

In leaves (Figure 4), soil TP correlated positively with P_i_ (*R*^2^ = 0.29, *p* = 0.04) and P_R_ (*R*^2^ = 0.32, *p* = 0.03) concentrations, whereas P_M_ and P_L_ followed unimodal curves that peaked at intermediate soil TP (P_M_: *R*^2^ = 0.73, *p* < 0.001; P_L_: *R*^2^ = 0.44, *p* = 0.03). No relationships were detected for leaf P_N_ or total leaf TP (P_N_: *R*^2^ = 0.01, *p* = 0.36; TP: *R*^2^ = 0.22, *p* = 0.09). The proportional data told a similar story: rP_i_ and rP_R_ increased linearly with soil TP (rP_i_: *R*^2^ = 0.44, *p* = 0.007; rP_R_: *R*^2^ = 0.28, *p* = 0.04), while rP_M_ and rP_N_ exhibited significant hump-shaped and U-shaped responses, respectively (*p* ≤ 0.03); rP_L_ remained unaffected (*p* = 0.10).

Below ground (Figure 4), soil TP was a strong linear predictor of root P_i_ (*R*^2^ = 0.87, *p* < 0.001), P_N_ (*R*^2^ = 0.60, *p* < 0.001), P_L_ (*R*^2^ = 0.30, *p* = 0.04), P_R_ (*R*^2^ = 0.44, *p* = 0.007) and root TP itself (*R*^2^ = 0.84, *p* < 0.001), but only a quadratic predictor of P_M_ (*R*^2^ = 0.43, *p* = 0.03). Allocation patterns shifted accordingly: rP_i_ increased with soil TP (*R*^2^ = 0.44, *p* = 0.007), whereas rP_N_, rP_L_ and rP_M_ all declined steeply (*R*^2^ ≥ 0.42, *p* ≤ 0.009). The relationship between rP_R_ and soil TP is not significant (*R*^2^ = 0.16, *p* = 0.14).

### 3.5. Correlations of P Fractions with Functional Traits in Leaves and Roots

Pore over the leaf data first (Figure 5A). Senesced leaf TP tracked green leaf TP, P_M_, P_L_ and rP_M_ tightly (r = 0.73–0.81, *p* ≤ 0.001) but was negatively related to rP_N_ (*r* = −0.58, *p* = 0.02). Consequently, PRE declined with increasing green leaf TP, P_M_, P_L_, rP_M_ and rP_L_ (*r* = −0.77 to −0.57, *p* ≤ 0.01) and increased only with rP_N_ (*r* = 0.52, *p* = 0.04). Both Pn and PPUE decreased sharply with higher green leaf TP, P_i_, P_M_, P_L_ and rP_M_ (*r* = 0.55–0.87, *p* ≤ 0.03) and increased with rP_N_ (*r* = −0.57 to −0.54, *p* ≤ 0.04). When leaf TP was set as a control variable (Table 2), partial correlation analysis found that senesced leaf TP was significantly positively correlated with the relative proportion of P_M_ (rP_M_), and correspondingly, PRE was significantly negatively correlated with rP_M_. However, leaf photosynthetic traits (Pn and PPUE) were not significantly correlated with the relative allocation of P fractions.

Root architecture was strongly linked to P chemistry (Figure 5B). SRL and SRA decreased with rising root TP, P_i_, P_L_, P_R_, rP_i_ and rP_R_ (*r* = −0.52 to −0.82, *p* ≤ 0.01) but increased with rP_N_ (*r* = 0.86 and 0.92, respectively, *p* < 0.001). RD showed the mirror image: negative correlations with TP, P_i_, P_N_, P_L_, P_R_ and rP_i_ (*r* = −0.74 to −0.92, *p* ≤ 0.001) and positive associations with rP_M_, rP_N_ and rP_L_ (*r* = 0.55–0.83, *p* ≤ 0.03). RTD increased with TP, P_i_, P_N_, P_L_, P_R_ and rP_i_ (*r* = 0.66–0.90, *p* ≤ 0.008) and decreased with rP_M_, rP_N_ and rP_L_ (*r* = −0.53 to −0.88, *p* ≤ 0.04). With root TP set as a control variable (Table 2), partial correlation analysis revealed that SRL and SRA were significantly positively correlated with rP_N_, while RD and RTD showed no significant relationships with the relative allocation of any P fractions.

## 4. Discussion

### 4.1. The Response of Leaf and Root P Fractions to Soil P Supply

Our study provides a critical case for resolving the persistent contradiction between crop studies, which often emphasize P reallocation as a key mechanism for maintaining photosynthetic efficiency under low P, and field observations in wild species [23]. Contrary to this widespread paradigm, our findings demonstrate that for *Y. shuichengensis* growing along a natural soil P gradient, the reallocation of P among biochemical fractions is not the primary mechanism sustaining photosynthetic rate. Instead, the absolute concentration of P in key functional pools appears decisive. This finding has direct implications for alpine bamboo ecosystem management, suggesting that maintaining soil P within an optimal range (e.g., ~1.2–1.6 g kg^−1^ in this study) to support adequate P uptake is more critical than selecting for internal reallocation traits in restoration practices.

The unimodal response of foliar P_M_ and P_L_, peaking at medium soil P levels, has traditionally been interpreted as an optimal metabolic investment. However, we propose a more nuanced interpretation that also considers passive processes [8]. The sharp increase in P_i_ and P_R_ at the high-P site could equally arise from a dilution effect at high soil P availability or represent an early detoxification mechanism against incipient P toxicity [24]. The preferential sequestration of P_i_ in vacuoles and P_R_ in cell walls is consistent with compartmental detoxification observed under luxury consumption. Thus, the observed pattern at the high-P site is likely the joint outcome of adaptive regulation (e.g., luxury storage) and passive processes (dilution/toxicity). Future work should combine leaf ATP, reactive oxygen assays and Nano-SIMS sub-cellular imaging to test toxicity thresholds explicitly [25].

Root P fractions exhibited a strong linear relationship with soil TP, supporting the ‘tight coupling’ theory between root chemistry and geochemical supply [26,27]. The nonlinear decrease in root P_M_ at very high soil TP suggests metabolic suppression and a shift towards surplus P storage, potentially signaling the onset of P saturation [24].

### 4.2. Differences in P Fraction Concentrations and Relative Allocations Between Leaves and Roots

A key finding is the fundamental difference in how leaves and roots manage P. Leaves maintained relatively stable P concentrations across the soil P gradient, demonstrating a strong capacity for P homeostasis, which is crucial for stabilizing photosynthetic function. In contrast, root P concentrations were more plastic, closely reflecting soil P availability. This organ-level specialization reflects their distinct physiological roles [15,23,25,28].

Leaves, as primary photosynthetic organs, prioritized maintaining higher concentrations of Pi, PN and PL, supporting photosynthetic protein synthesis, membrane integrity and energy metabolism. Roots, as acquisition organs, showed allocation patterns more directly tied to soil P supply, shifting towards a ‘low-cost, high-throughput’ strategy with increased Pi storage under high P conditions. The positive correlation between root PN allocation and root elongation underscores the role of nucleic acid P in sustaining growth under variable P supply [27,29]. The “high concentration–low reallocation” strategy may enhance the resilience of *Y. shuichengensis* to environmental fluctuations, supporting rapid photosynthetic recovery in early spring and contributing to ecosystem stability under climate change.

### 4.3. Effects of Plant P Fractions on Functional Traits

The relationship between P fractions and key functional traits further underscores that P concentration, not reallocation, is the dominant factor for *Y. shuichengensis* in its natural environment. Photosynthetic performance (Pn and PPUE) was driven by the absolute concentrations of P fractions (especially P_M_ and P_L_) rather than by their relative proportions [11,12,30]. This suggests that this bamboo species enhances photosynthesis primarily by increasing the overall P pool in metabolic tissues under favorable conditions, rather than by reshuffling internal P allocations under stress.

During leaf senescence, PRE declined with increasing green leaf TP, P_M_, and P_L_, indicating that metabolic P pools may be preferentially abandoned [24,31,32]. This implies that PRE is governed more by the overall P status of the leaf than by active, fraction-specific resorption mechanisms, a finding that aligns with some field studies but contrasts with patterns observed in some fertilized systems [31,32].

Belowground, root morphology was closely linked to P chemistry [29,33]. The shift from a thick-root, mycorrhiza-mediated strategy under low P to a thin-root, direct-uptake strategy under high P reflects an adaptive transition in P acquisition mode, which is efficiently supported by changes in P concentration rather than reallocation. Future research should couple P fractionation with direct measurements of physiological stress and carbon fluxes along climate-elevation gradients to quantify how this P-use strategy contributes to carbon sequestration and overall ecosystem functioning.

In conclusion, our study refines the understanding of plant P allocation by demonstrating that in a wild bamboo species under natural conditions, the absolute concentration of P in functional pools supersedes internal reallocation as the primary driver of physiological performance. This context-dependent outcome calls for a more nuanced paradigm that incorporates species-specific strategies and environmental gradients beyond the model derived from crop studies.

## 5. Conclusions

This study systematically quantified the P fractions and their allocation in *Y. shuichengensis* along a natural soil P gradient. In leaves, P_M_ and P_L_ peaked at moderate P supply, while high-P environments shifted toward P_i_ and P_R_ storage. In roots, P_i_ increased linearly with soil TP, reflecting tight coupling to geochemical supply. Leaf P_i_, P_N_ and P_L_ concentrations were significantly higher than in roots, underscoring the role of leaves in photosynthetic homeostasis and roots in storage and buffering. Leaf P fractions often exhibited nonlinear responses with thresholds, whereas root P fractions followed largely linear trends. Leaf PRE was driven by leaf TP and the absolute concentrations of P_M_ and P_L_, but not by their relative allocation. This indicates that for *Y. shuichengensis* in its natural habitat, P fraction reallocation may not be a primary mechanism for enhancing photosynthesis or resorption efficiency, highlighting the importance of absolute P content. After controlling root TP, SRL and SRA remained positively correlated with rP_N_, supporting a role for P_N_ in promoting root elongation and acquisition.

This study has limitations, as findings are based on a single species along a natural gradient. Generalization to other species or environments requires caution. Future work should expand to include a wider phylogenetic range and experimental manipulations to test the generality of these P allocation strategies. Research into the molecular regulation of P fraction partitioning and its integration with whole-plant carbon balance will further elucidate the ecological significance of these patterns.

In summary, *Y. shuichengensis* displays organ- and fraction-specific P allocation strategies across soil P gradients. The prioritization of concentration regulation over internal reallocation under natural conditions likely supports its adaptation and may influence the spatial distribution of high-altitude bamboos in heterogeneous landscapes.

## Figures and Tables

**Figure 1 biology-14-01647-f001:**
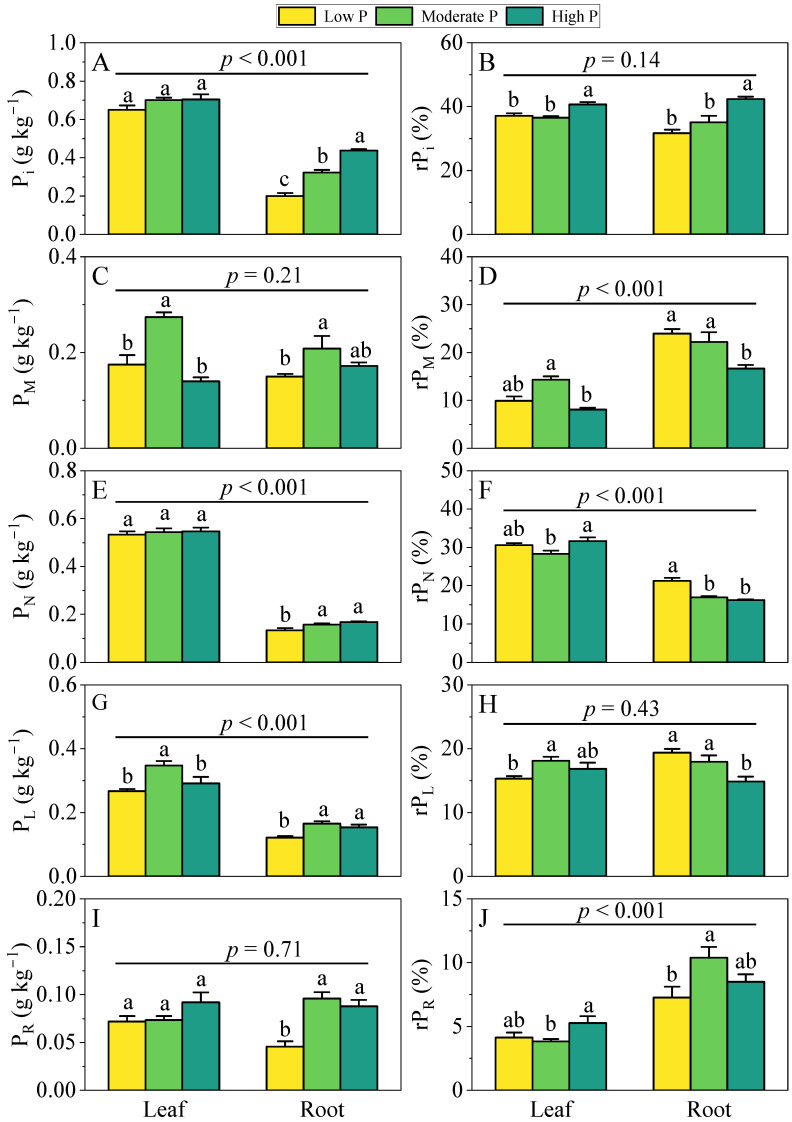
Differences in concentrations and relative allocations of five P fractions in leaves and roots of *Y. shuichengensis* across soil P concentration gradients (low-P, moderate-P and high-P sites). (**A**), orthophosphate P (P_i_) concentration; (**B**), relative allocation of P_i_ (rP_i_); (**C**), low-molecular-weight metabolite P (P_M_) concentration; (**D**), relative allocation of P_M_ (rP_M_); (**E**), nucleic acid P (P_N_) concentration; (**F**), relative allocation of P_N_ (rP_N_); (**G**), lipid P (P_L_) concentration; (**H**), relative allocation of P_L_ (rP_L_); (**I**), residual P (P_R_) concentration; (**J**), relative allocation of P_R_ (rP_R_). Different lowercase letters indicate significant differences among different lithology in the same plant organ (*p* < 0.05). The *p*-values above the line are the results of paired *t*-tests for differences in P fraction concentrations and allocation between leaves and roots. Values are means ± SE.

**Figure 2 biology-14-01647-f002:**
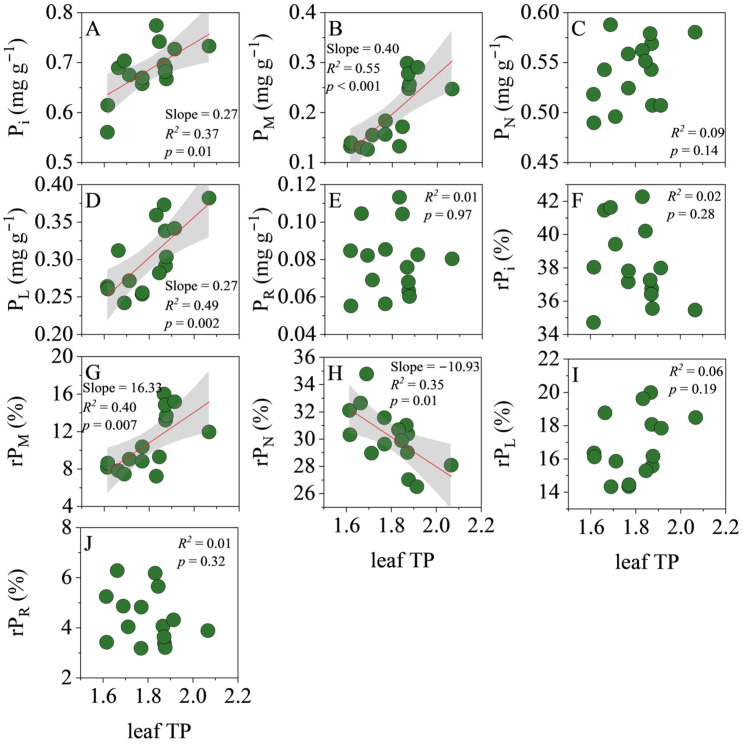
Linear regression analysis of the total P (TP) concentration with the concentration and relative allocation of each P fraction in leaves. (**A**), orthophosphate P (P_i_) concentration; (**B**), low-molecular-weight metabolite P (P_M_) concentration; (**C**), nucleic acid P (P_N_) concentration; (**D**), lipid P (P_L_) concentration; (**E**), residual P (P_R_) concentration; (**F**), relative allocation of P_i_ (rP_i_); (**G**), relative allocation of P_M_ (rP_M_); (**H**), relative allocation of P_N_ (rP_N_); (**I**), relative allocation of P_L_ (rP_L_); (**J**), relative allocation of P_R_ (rP_R_). The shaded area represents the 95% confidence interval.

**Figure 3 biology-14-01647-f003:**
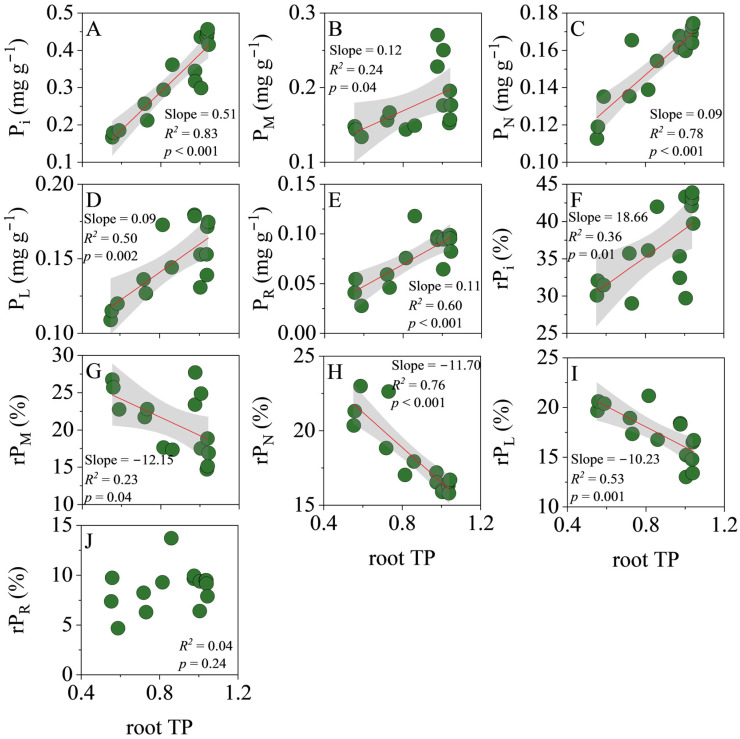
Linear regression analysis of the total P (TP) concentration with the concentration and relative allocation of each P fraction in roots. (**A**), orthophosphate P (P_i_) concentration; (**B**), low-molecular-weight metabolite P (P_M_) concentration; (**C**), nucleic acid P (P_N_) concentration; (**D**), lipid P (P_L_) concentration; (**E**), residual P (P_R_) concentration; (**F**), relative allocation of P_i_ (rP_i_); (**G**), relative allocation of P_M_ (rP_M_); (**H**), relative allocation of P_N_ (rP_N_); (**I**), relative allocation of P_L_ (rP_L_); (**J**), relative allocation of P_R_ (rP_R_). The shaded area represents the 95% confidence interval.

**Figure 4 biology-14-01647-f004:**
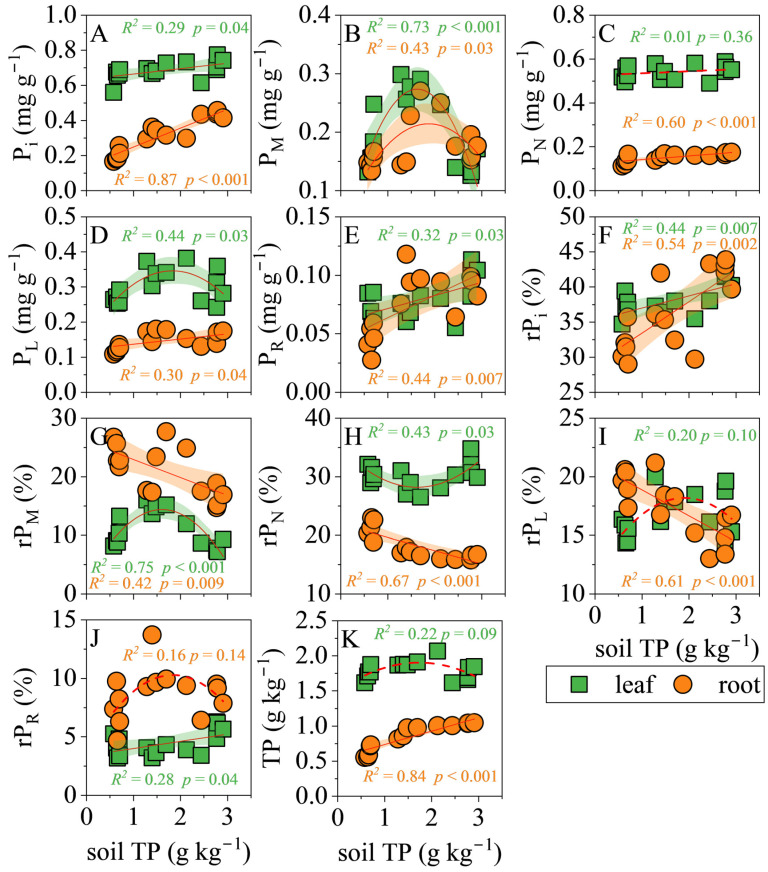
Linear regression analysis of the soil total P (TP) concentration with the concentration and relative allocation of each P fraction in leaves and roots. (**A**), orthophosphate P (P_i_) concentration; (**B**), low-molecular-weight metabolite P (P_M_) concentration; (**C**), nucleic acid P (P_N_) concentration; (**D**), lipid P (P_L_) concentration; (**E**), residual P (P_R_) concentration; (**F**), relative allocation of P_i_ (rP_i_); (**G**), relative allocation of P_M_ (rP_M_); (**H**), relative allocation of P_N_ (rP_N_); (**I**), relative allocation of P_L_ (rP_L_); (**J**), relative allocation of P_R_ (rP_R_); (**K**), TP in leaf or root. The shaded area represents the 95% confidence interval.

**Figure 5 biology-14-01647-f005:**
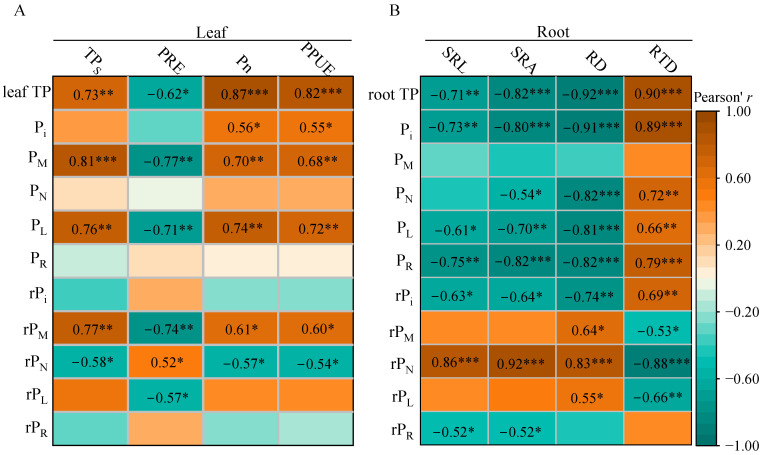
Pearson correlation analysis between the concentrations and relative allocations of P fractions and leaf functional traits (**A**) or root functional traits (**B**). TP, total P; TP_S_, senesced leaf TP; PRE, P resorption efficiency; Pn, net photosynthetic rate; PPUE, photosynthetic P-use efficiency; SRL, specific root length; SRA, specific root surface area; RD, root diameter; RTD, root tissue density; P_i_, orthophosphate P concentration; P_M_, low-molecular-weight metabolite P concentration; P_N_, nucleic acid P concentration; P_L_, lipid P concentration; P_R_, residual P concentration; rP_i_, relative allocation of P_i_; rP_M_, relative allocation of P_M_; rP_N_, relative allocation of P_N_; rP_L_, relative allocation of P_L_; rP_R_, relative allocation of P_R_. Significance level: *, 0.01 ≤ *p* < 0.05; **, 0.001 ≤ *p* < 0.01; ***, *p* < 0.001.

**Table 1 biology-14-01647-t001:** Description of the three study sites. Soil total phosphorus (TP) represents the concentration of TP in 0–20 cm soil.

Site Name	Low-P Site	Moderate-P Site	High-P Site
Coordinate	26°50′40″ N, 104°41′29″ E	26°50′30″ N, 104°42′10″ E	27°0′25″ N, 104°43′12″ E
Parent rock	Sandstone	Limestone	Basalt
Vegetation type	Bamboo shrubland	Bamboo shrubland	Bamboo shrubland
MAT (°C)	8.1	8.1	8.0
MAP (mm yr^−1^)	1094	1094	1088
Soil TP (g kg^−1^)	0.66	1.60	2.73

**Table 2 biology-14-01647-t002:** Partial correlation analysis between the relative allocations of P fractions and leaf functional traits or root functional traits. TP, total P; TP_S_, senesced leaf TP; PRE, P resorption efficiency; Pn, net photosynthetic rate; PPUE, photosynthetic P-use efficiency; SRL, specific root length; SRA, specific root surface area; RD, root diameter; RTD, root tissue density; rP_i_, relative allocation of orthophosphate P; rP_M_, relative allocation of low-molecular-weight metabolite P; rP_N_, relative allocation of nucleic acid P; rP_L_, relative allocation of lipid P; rP_R_, relative allocation of residual P. Significance level: *, 0.01 ≤ *p* < 0.05; **, 0.001 ≤ *p* < 0.01.

Organ	Control Variable	Functional Traits	rP_i_	rP_M_	rP_N_	rP_L_	rP_R_
Leaf	leaf TP	TP_S_	−0.18	0.56 *	−0.23	0.48	−0.14
		PRE	0.16	−0.57 *	0.22	−0.48	0.14
		Pn	0.02	0.09	−0.04	0.31	0.08
		PPUE	0.09	0.13	−0.03	0.31	0.09
Root	root TP	SRL	−0.32	0.07	0.71 **	−0.26	−0.44
		SRA	−0.26	0.06	0.72 **	−0.34	−0.48
		RD	−0.51	0.45	0.09	−0.53	−0.37
		RTD	0.35	−0.13	−0.41	0.05	0.30

## Data Availability

Data will be made available on request.

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
