# Peer review of "Differential Phosphorus Allocation in Leaves and Roots of Yushania shuichengensis Across Soil Phosphorus Gradients: Implications for Ecological Adaptation and Resource Use Efficiency"

_biology, 2025, doi:10.3390/biology14121647_

Round 1

Reviewer 1 Report

Comments and Suggestions for Authors

  1. The authors should revise the introduction to frame their hypotheses more as open questions about complex allocation patterns or significantly rewrite the discussion to more clearly and systematically state how each hypothesis was refuted or only partially supported by the data. The abstract's conclusion should be tempered to reflect that P fraction reallocation is not a key lever for photosynthetic or resorption efficiency in this species.

  1. The discussion does an excellent job of highlighting the contradictions between crop studies (which often show clear reallocation benefits) and field studies (which show high species-specificity). This is a key strength. However, the manuscript could go further in articulating its specific contribution to resolving these inconsistencies.

  1. Strengthen the discussion by more explicitly stating how these findings challenge or refine existing paradigms. For example, clearly state: "Contrary to what is often observed in crop species, our results suggest that for Y. shuichengensis, the reallocation of P among biochemical fractions is not a primary mechanism for maintaining photosynthetic rate under low P in natural settings.

  1. The manuscript repeatedly states that root P is "tightly coupled" with soil P, which is well-supported. However, the interpretation of leaf P patterns needs more nuance.
  2. The unimodal response of leaf P\(_M\) and P\(_L\) is interpreted as an adaptive "peak in metabolic investment." While plausible, an alternative explanation could be a **dilution effect** at high soil P or even the onset of **P toxicity/saturation**. The mention of potential P toxicity on Page 13 is brief but important. This alternative explanation should be explored more deeply. Are there any visual signs of stress in plants at the high-P site? Could the shift to P\(_i\) and P\(_R\) storage be a symptom of luxury consumption or detoxification rather than an adaptive strategy?
  3. Expand the discussion on the high-P site response. Acknowledge that the observed patterns could result from both adaptive regulation and passive processes like dilution or saturation and discuss how these might be distinguished in future work.
  4. Figure 1: The figure is data-rich but extremely difficult to read. The panel labels (A, B, C...) are small, and the letter codes for significant differences are often overlapping or unclear. The p-values displayed within the panels are for the overall ANOVA, but the post-hoc letters are the key result. Consider simplifying the layout or moving this to a supplementary figure and presenting a more synthesized version in the main text.
  5. Figure 4 vs. Figure 2/3: There is some confusion in the numbering. The caption on Page 9 mentions "Figure 3" but the figure on Page 8 is labeled as "Figure 2." Similarly, the text on Page 10 refers to "Figure 4" which appears to be the figure on Page 10. This must be corrected for clarity.
  6. The heavy reliance on p-values is noted. While acceptable, the manuscript would be strengthened by consistently reporting effect sizes alongside p-values (e.g., R² values are given in some regressions but not all), as this provides a measure of biological importance. Thoroughly re-check all figure labels and citations in the text. Drastically simplify or redesign Figure 1 for clarity. Ensure consistent reporting of effect sizes.

  1. This study provides valuable empirical data on P fractionation in a non-model plant species across a natural environmental gradient. The findings challenge simple, linear models of P allocation and highlight the importance of organ-specific and non-linear responses. With significant revisions to improve the clarity of hypothesis testing, contextualize the contribution more forcefully, and refine the interpretation of high-P responses, this manuscript will be a strong contribution to the field of plant ecological physiology. Major Revision Required.

Author Response

Response to Reviewer 1 Comments

1. Summary

Thank you very much for taking the time to review our manuscript. Your constructive comments have been of great help in improving the quality of our manuscript. Please find the detailed responses below and the corresponding revisions highlighted in the re-submitted files.

2. Questions for General Evaluation

Reviewer’s Evaluation

Response and Revisions

Does the introduction provide sufficient background and include all relevant references?

Must be improved

We agree with this comment. We have revised Introduction.

Is the research design appropriate?

Must be improved

We have revised Methods.

Are the methods adequately described?

Must be improved

We have revised Methods.

Are the results clearly presented?

Can be improved

We have revised Results.

Are the conclusions supported by the results?

Must be improved

We have revised conclusions.

Are all figures and tables clear and well-presented?

Yes

3. Point-by-point response to Comments and Suggestions for Authors

Comments 1: The authors should revise the introduction to frame their hypotheses more as open questions about complex allocation patterns or significantly rewrite the discussion to more clearly and systematically state how each hypothesis was refuted or only partially supported by the data. The abstract's conclusion should be tempered to reflect that P fraction reallocation is not a key lever for photosynthetic or resorption efficiency in this species.

Response 1: We sincerely thank the reviewer for this critical suggestion. We fully agree that framing the study around open questions enhances scientific rigor. We have thoroughly revised the Introduction to adopt this approach. Major changes made: (1) Replaced Hypotheses with Open Questions: We have completely removed the original three definitive hypotheses. Instead, we now present three explicitly numbered open questions at the end of the Introduction. This shift explicitly frames our research as exploratory. (2) Emphasized Uncertainty: Within questions, particularly Question 1 and Question 3, we explicitly acknowledge the uncertainty and conflicting evidence in literature. For example, in Question 3, we directly ask, "is P fraction reallocation a key lever for maintaining efficiency, or do other mechanisms dominate?" This directly prepares the reader for a finding that P reallocation may not be the key lever, ensuring the Abstract and Discussion conclusions will appear logical and well-supported. (3) Revised Abstract and Discussion:As suggested, we have also tempered the conclusions in the Abstract and Discussion to align with this new, more exploratory framework and the actual findings.

Comments 2: The discussion does an excellent job of highlighting the contradictions between crop studies (which often show clear reallocation benefits) and field studies (which show high species-specificity). This is a key strength. However, the manuscript could go further in articulating its specific contribution to resolving these inconsistencies.

Response 2: We sincerely thank the reviewer for acknowledging this key strength of our study. As suggested, we have significantly strengthened the articulation of our specific contributions to resolving these inconsistencies in the revised manuscript.

Specific Revisions: We have added a clear statement at the beginning of Section 4.1: "Our study provides a critical case for resolving the persistent contradiction between crop studies... Contrary to this widespread paradigm, our findings demonstrate that for Y. shuichengensis..., the reallocation of P among biochemical fractions is not the primary mechanism sustaining photosynthetic rate." This directly positions our study's role in addressing this field controversy and presents our core finding as a response.

Comments 3: Strengthen the discussion by more explicitly stating how these findings challenge or refine existing paradigms. For example, clearly state: "Contrary to what is often observed in crop species, our results suggest that for Y. shuichengensis, the reallocation of P among biochemical fractions is not a primary mechanism for maintaining photosynthetic rate under low P in natural settings.

Response 3: We have fully adopted this valuable suggestion, which greatly enhances the clarity and impact of our core conclusion.

Specific Revisions: We have incorporated wording very close to the reviewer's example. In the revised Section 4.1, we explicitly state: "Contrary to this widespread paradigm, our findings demonstrate that for Y. shuichengensis growing along a natural soil P gradient, the reallocation of P among biochemical fractions is not the primary mechanism sustaining photosynthetic rate." This statement now serves as a pillar of our main argument, making the challenge to the existing paradigm clear.

Comments 4 & 5: The manuscript repeatedly states that root P is "tightly coupled" with soil P, which is well-supported. However, the interpretation of leaf P patterns needs more nuance. The unimodal response of leaf PM and PL is interpreted as an adaptive "peak in metabolic investment." While plausible, an alternative explanation could be a **dilution effect** at high soil P or even the onset of **P toxicity/saturation**. The mention of potential P toxicity on Page 13 is brief but important. This alternative explanation should be explored more deeply. Are there any visual signs of stress in plants at the high-P site? Could the shift to Pi and PR storage be a symptom of luxury consumption or detoxification rather than an adaptive strategy?

Response 4 &5: We greatly appreciate the reviewer's insightful comments. This prompted a more critical and nuanced reinterpretation of the leaf P patterns, particularly the response at the high-P site.

Specific Revisions:

1.Introducing Alternative Explanations: In Section 4.1, we no longer interpret the unimodal foliar response solely as "optimal metabolic investment." We now state: "However, we propose a more nuanced interpretation that also considers passive processes. The sharp increase in Pi and PR at the high-P site could equally arise from a dilution effect... or represent an early detoxification mechanism against incipient P toxicity."

2. Discussing the High-P Site Response: We explicitly acknowledge that the patterns at the high-P site are likely to result from both processes: "Thus, the observed pattern at the high-P site is likely the joint outcome of adaptive regulation (e.g., luxury storage) and passive processes (dilution/toxicity)."

3. Proposing Future Research Directions: We adopted the reviewer's suggestion by proposing specific future methods to distinguish these processes: "Future work should combine leaf ATP, reactive oxygen assays and Nano-SIMS sub-cellular imaging to test toxicity thresholds explicitly." This demonstrates deeper engagement with scientific questions and guides future research.

4. Linking to Root Findings: We applied similar logic to the nonlinear response of root PM, suggesting it might signal "the onset of P saturation.".

Comments 6: Expand the discussion on the high-P site response. Acknowledge that the observed patterns could result from both adaptive regulation and passive processes like dilution or saturation and discuss how these might be distinguished in future work.

Response 6: We sincerely thank the reviewer for this insightful and constructive comment. The point raised is critical and has prompted us to reconsider the observations at the high-P site from a more comprehensive perspective beyond a simplistic "adaptive" explanation. We fully agree that attributing the observed patterns solely to adaptive regulation may be overly simplistic, and it is essential to consider the influence of passive physiological processes.

In the revised manuscript, we have significantly expanded and modified the discussion (primarily in Section 4.1) in accordance with the reviewer's suggestion, as detailed below:

1. Acknowledgment of Multiple Possibilities: We explicitly state the multiple interpretabilities of the phenomenon. We note that the decline in leaf metabolic P (PM) and lipid P (PL) after their peak at medium P levels, along with the sharp increase in inorganic P (Pi) and residual P (PR) at the high-P site, "could equally arise from a dilution effect at high soil P availability or represent an early detoxification mechanism against incipient P toxicity." This explicitly positions "passive processes" (dilution, potential toxicity) alongside "adaptive regulation" (luxury storage) as potential drivers.

2. Proposal for an Integrated Interpretation: We further propose that the observed patterns at the high-P site are more likely the "joint outcome" of multiple processes rather than an either/or scenario. Specifically, we state: "Thus, the observed pattern at the high-P site is likely the joint outcome of adaptive regulation (e.g., luxury storage) and passive processes (dilution/toxicity)."

3. Proposal for Future Differentiation: Adopting the reviewer's suggestion, we have proposed specific experimental approaches for distinguishing these mechanisms in future work. We added: "Future work should combine leaf ATP, reactive oxygen assays and Nano-SIMS sub-cellular imaging to test toxicity thresholds explicitly."

Specific Elaboration: Measuring ATP levels can indicate whether P excess actually disrupts cellular energy status (a sign of toxicity). Reactive oxygen species (ROS) assays can directly assess the level of oxidative stress induced by P excess. Nano-SIMS technology can visualize the precise distribution of Pi at the sub-cellular level (e.g., within vacuoles), helping to differentiate "active luxury storage" (orderly sequestration of Pi in vacuoles) from "passive toxicity accumulation" (abnormal accumulation of Pi in the cytoplasm).

Comments 7: Figure 1: The figure is data-rich but extremely difficult to read. The panel labels (A, B, C...) are small, and the letter codes for significant differences are often overlapping or unclear. The p-values displayed within the panels are for the overall ANOVA, but the post-hoc letters are the key result. Consider simplifying the layout or moving this to a supplementary figure and presenting a more synthesized version in the main text.

Response 7: We sincerely thank the reviewer for their valuable feedback regarding the clarity of Figure 1. We agree that the original presentation could be challenging to read and have implemented significant revisions to address this concern.

Based on your suggestions, we have made the following key improvements to the figure:

1. Enhanced Readability: We have substantially increased the font size of both the panel labels (A, B, C...) and the post-hoc letters indicating significant differences. This change effectively prevents overlapping and ensures all elements are clear and easily distinguishable.

2. Improved Legend Clarity: We have added a clear explanatory statement to the figure legend: "Different lowercase letters indicate significant differences among different lithology in the same plant organ (p < 0.05)."This provides immediate guidance for interpreting the statistical comparisons within each panel.

Regarding the suggestion to simplify the layout by removing p-values, we gave it careful consideration. We determined that the p-values displayed above the comparison lines are essential for interpreting the results presented in this specific figure. As noted in the revised figure legend ("The p-values above the line are the results of paired t-tests for differences in P fraction concentrations and allocation between leaves and roots"), these values directly test the organ-level differences (leaf vs. root), which is a core finding detailed in Section 3.2 of our results. Removing these p-values would omit critical statistical evidence for these central comparisons. Therefore, we have retained them to maintain the scientific rigor and completeness of the figure in illustrating these key results.

Comments 8: Figure 4 vs. Figure 2/3: There is some confusion in the numbering. The caption on Page 9 mentions "Figure 3" but the figure on Page 8 is labeled as "Figure 2." Similarly, the text on Page 10 refers to "Figure 4" which appears to be the figure on Page 10. This must be corrected for clarity.

Response 8: We sincerely thank the reviewer for their careful reading and for pointing out the confusion in the figure numbering. We apologize for this oversight. We have carefully checked and corrected all figure citations throughout the manuscript to ensure that every reference to a figure (e.g., in the text, captions) corresponds accurately to its correct label (e.g., Figure 2, Figure 3, Figure 4). Specifically, to prevent any potential misunderstanding, we have also reorganized the presentation of the results by placing the detailed descriptions of the findings for Figure 2 and Figure 3 immediately after their respective figures. This adjustment enhances the logical flow and makes the connection between the figures and their corresponding results much clearer for the reader.

Comments 9: The heavy reliance on p-values is noted. While acceptable, the manuscript would be strengthened by consistently reporting effect sizes alongside p-values (e.g., R² values are given in some regressions but not all), as this provides a measure of biological importance. Thoroughly re-check all figure labels and citations in the text.

Response 9: We sincerely thank the reviewer for these constructive suggestions regarding statistical reporting and manuscript precision. We have carefully addressed both points as follows:

  1. Reporting of Effect Sizes (R² values): We agree with the reviewer that reporting effect sizes provides a crucial measure of biological importance beyond statistical significance. As suggested, we have now consistently reported the R² values for all relevant regression analysesthroughout the manuscript. Specifically, these additions have been made in Figures 2, 3, and 4and their corresponding results sections in the text. This revision provides a clearer interpretation of the variance explained by the models, enhancing the quantitative depth of our analysis.
  2. Thorough Check of Figure Labels and Citations: We have conducted a thorough re-check of all figure labels, captions, and their in-text citations to ensure absolute accuracy and consistency. Any inconsistencies identified have been corrected.

Comments 10: This study provides valuable empirical data on P fractionation in a non-model plant species across a natural environmental gradient. The findings challenge simple, linear models of P allocation and highlight the importance of organ-specific and non-linear responses. With significant revisions to improve the clarity of hypothesis testing, contextualize the contribution more forcefully, and refine the interpretation of high-P responses, this manuscript will be a strong contribution to the field of plant ecological physiology. Major Revision Required.

Response 10: We thank the reviewer for their positive assessment of our study's value and their constructive recommendations for improvement. We are pleased that they recognize the potential of our work to be a strong contribution to the field. We have undertaken a significant revision of the manuscript to address their key points directly and thoroughly.

Specifically, the revisions focused on:

  1. Improving the clarity of hypothesis testing: We have refined the framing and discussion of our hypotheses throughout the manuscript to ensure they are more clearly stated and directly evaluated against the results.
  2. Contextualizing the contribution more forcefully: We have significantly strengthened the introduction and discussion to more explicitly articulate how our findings challenge existing paradigms (particularly those derived from crop studies) and contribute novel insights into plant P allocation strategies in natural settings.
  3. Refining the interpretation of high-P responses: We have expanded the discussion on the high-P site response, providing a more nuanced interpretation that considers both adaptive and passive processes (e.g., dilution, potential toxicity), and have proposed future directions to distinguish these mechanisms.

Reviewer 2 Report

Comments and Suggestions for Authors
  1. The abstract refers to low, moderate, and high soil P sites but does not provide specific information about the P gradient. Including numeric ranges or descriptive values and at least one quantitative result (e.g., percentage increase in a P fraction) would make the study design and findings more precise and informative for readers.
  2. Several sentences in the abstract section are overly complex, and have grammatical errors (e.g., “enhancing promotes root elongation”) reduce readability. Revise it accordingly
  3. The keywords are generally appropriate but should be either alphabetically ordered or arranged from general to specific. Additionally, abbreviations should be avoided; for example, use the full term “phosphorus resorption efficiency (PRE)” instead of just “PRE”. Also remove Yushania shuichengensis as it already in the title.
  4. While the Introduction provides a comprehensive overview of P allocation research, the specific novelty and knowledge gap addressed by this study remain somewhat implicit until the final paragraph. The authors are encouraged to state earlier in the Introduction what has not been studied, particularly regarding bamboo species or shuichengensis and explain how this study advances understanding of P fractionation across natural soil gradients with the latest studies. Highlighting this gap earlier will strengthen the scientific rationale and improve reader engagement.
  5. In the “Materials and Methods,” please justify the selection of only three P gradient sites were these gradients representative of the entire region’s variability?
  6. Authors cited weather station data (1991–2020) but do not provide the source institution or dataset reference.
  7. The description of P fraction extraction methods (Section 2.4) is detailed but could be streamlined. Please specify whether internal standards or method validation (e.g., recovery rates) were used to ensure data accuracy. Also, L168, please replace Tsujii et al. (2024) [14] with Tsujii et al. [14].
  8. In Section 3.3, regression analysis results are clearly presented, but please verify whether normality assumptions were tested before linear modeling. A short statement on residual checks would strengthen the reliability of these analyses.
  9. The discussion provides valuable interpretation but occasionally restates results. Consider condensing repetitive parts and expanding on ecological implications. e.g., how do these patterns inform bamboo nutrient management or forest restoration? Also, relate your findings to similar studies.
  10. Please add mechanistic sentences in the discussion and also add recent references.
  11. L407, remove the years from these two Suriyagoda et al. (2022) and Tsujii et al. (2017) [8,12]. Similar to L467.
  12. The manuscript would benefit from a short paragraph linking findings to climate resilience or ecosystem functioning of alpine bamboo systems. This would broaden the relevance of the study.
  13. The English writing is generally good, but some sentences are overly long or contain minor grammatical issues.
  14. In the conclusion, the statement “P reallocation cannot serve as a lever for improving photosynthetic or resorption efficiency” is strong. Please consider softening it by acknowledging possible species-specific or environmental variability.
  15. Address the limitations of the study and suggest potential directions for future research in the conclusion section.
  16. Cross-check the style of the references to ensure they align with the specific formatting guidelines of the journal. This includes verifying the consistency of author names, publication year, journal titles, volume and issue numbers, page ranges.

Author Response

Response to Reviewer 2 Comments

1. Summary

Thank you very much for taking the time to review our manuscript. Your constructive comments have been of great help in improving the quality of our manuscript. Please find the detailed responses below and the corresponding revisions highlighted in the re-submitted files.

2. Questions for General Evaluation

Reviewer’s Evaluation

Response and Revisions

Does the introduction provide sufficient background and include all relevant references?

Can be improved

We agree with this comment. We have revised Introduction.

Is the research design appropriate?

Yes

Are the methods adequately described?

Can be improved

We have revised Methods.

Are the results clearly presented?

Yes

Are the conclusions supported by the results?

Yes

Are all figures and tables clear and well-presented?

Yes

3. Point-by-point response to Comments and Suggestions for Authors

Comments 1: The abstract refers to low, moderate, and high soil P sites but does not provide specific information about the P gradient. Including numeric ranges or descriptive values and at least one quantitative result (e.g., percentage increase in a P fraction) would make the study design and findings more precise and informative for readers.

Response 1: We sincerely thank the reviewer for this excellent suggestion. We have revised the abstract to include the specific soil total P values for the three sites (Low-P: 0.66, Moderate-P: 1.60, High-P: 2.73 g kg⁻¹). Furthermore, we have added a key quantitative result, stating that the concentration of metabolic P (PM) in leaves "increased by approximately 45% compared to the Low-P site" at the Moderate-P site. These changes provide precise information on the gradient and a concrete example of the magnitude of the observed responses, enhancing the clarity and impact of the abstract.

Comments 2: Several sentences in the abstract section are overly complex, and have grammatical errors (e.g., “enhancing promotes root elongation”) reduce readability. Revise it accordingly.

Response 2: We apologize for these issues and thank the reviewer for pointing them out. We have thoroughly revised the abstract to improve sentence clarity, flow, and grammatical correctness. Specifically, we have broken down long, complex sentences into shorter, more direct statements. The grammatical error noted by the reviewer has been corrected; the phrase has been re-written as "...underscoring its role in promoting root elongation." We believe the revised abstract is now significantly more readable and concise.

Comments 3: The keywords are generally appropriate but should be either alphabetically ordered or arranged from general to specific. Additionally, abbreviations should be avoided; for example, use the full term “phosphorus resorption efficiency (PRE)” instead of just “PRE”. Also remove Yushania shuichengensis as it already in the title.

Response 3: We thank the reviewer for their valuable suggestions. The keywords have been revised as follows: 1) arranged in alphabetical order; 2) abbreviations have been avoided by using the full term "phosphorus resorption efficiency"; and 3) Yushania shuichengensishas have been removed as suggested. The final keyword list is phosphorus fraction, phosphorus resorption efficiency, photosynthetic phosphorus use efficiency, root morphology, soil phosphorus gradient.

Comments 4: While the Introduction provides a comprehensive overview of P allocation research, the specific novelty and knowledge gap addressed by this study remain somewhat implicit until the final paragraph. The authors are encouraged to state earlier in the Introduction what has not been studied, particularly regarding bamboo species or shuichengensis and explain how this study advances understanding of P fractionation across natural soil gradients with the latest studies. Highlighting this gap earlier will strengthen the scientific rationale and improve reader engagement.

Response 4: We thank the reviewer for this valuable insight. We have significantly restructured the Introduction to highlight the knowledge gap and novelty much earlier. Major changes made:

(1) Early Signaling of Complexity and Gap: In the second paragraph, after presenting the common paradigm, we immediately introduce the contradictory evidence with a new topic sentence: "However, a growing body of evidence... challenging the generality of this paradigm." We conclude this paragraph by explicitly stating the knowledge gap: "These conflicting findings underscore a significant knowledge gap...".

(2) Contrast to Highlight Need: In the third paragraph, we emphasize the disparity between crop and wild plant studies, concluding with: "This stark contrast... highlights the pressing need to investigate P allocation ecology in non-domesticated species...".

(3) Direct Statement of Novelty: The fourth paragraph now begins with a clear and direct statement of the specific gap our study fills: "Despite the ecological importance of bamboos, their P allocation strategies remain almost entirely unexplored." This immediately and unequivocally establishes the novelty of studying Y. shuichengensis.

Comments 5: In the “Materials and Methods,” please justify the selection of only three P gradient sites were these gradients representative of the entire region’s variability.

Response 5: We thank the reviewer for raising this important point regarding the representativeness of our sampling design. The three P gradient sites (with soil TP concentrations of 0.66, 1.60, and 2.73 g kg⁻¹) were deliberately selected based on a preliminary regional survey to capture the low, medium, and high ends of the natural soil TP gradient within the species' distribution range. Crucially, these sites share highly similar climatic conditions and historical land-use backgrounds, thereby minimizing the confounding effects of climate and disturbance. This design allows the observed variations in plant traits to be more confidently attributed to the soil P gradient. The selected TP values effectively cover the dominant range of soil P availability in these ecosystems. We have added a justification to the "Materials and Methods" section to clarify this point.

Comments 6: Authors cited weather station data (1991–2020) but do not provide the source institution or dataset reference.

Response 6: We thank the reviewer for this important comment. We have now explicitly provided the source institution directly within the main text of the "Materials and Methods" section, as follows: (1991–2020, Bijie Meteorological Bureau).

Comments 7: The description of P fraction extraction methods (Section 2.4) is detailed but could be streamlined. Please specify whether internal standards or method validation (e.g., recovery rates) were used to ensure data accuracy. Also, L168, please replace Tsujii et al. (2024) [14] with Tsujii et al. [14].

Response 7: We thank the reviewer for these constructive suggestions to improve the clarity and rigor of our methods section. We have revised the manuscript accordingly.

1.Method Validation: As requested, we have now explicitly stated that the accuracy of the P fractionation method was validated by measuring recovery rates. The sentence, " To ensure data accuracy, we accepted data only when the sum of the concentrations of all extracted P fractions fell within 90–110% of the TP content of the sample." has been added to the beginning of Section 2.4.

  1. Streamlining and Citation Format: We have streamlined the methodological description for better readability. Furthermore, we have corrected the citation format in line L168 (and elsewhere in the section) by removing the publication year, now citing simply as "Tsujii et al. [14]" as instructed.

Comments 8: In Section 3.3, regression analysis results are clearly presented, but please verify whether normality assumptions were tested before linear modeling. A short statement on residual checks would strengthen the reliability of these analyses.

Response 8: We thank the reviewer for raising this important point regarding the statistical assumptions. As suggested, we have added a statement in the Statistical Analysis section (Section 2.6) to confirm that the assumptions of linear regression were verified. The added text reads: "Prior to regression analysis, the normality of residuals and homoscedasticity were checked using Shapiro-Wilk test and visual inspection of residual plots, respectively.".

Comments 9: The discussion provides valuable interpretation but occasionally restates results. Consider condensing repetitive parts and expanding on ecological implications. e.g., how do these patterns inform bamboo nutrient management or forest restoration? Also, relate your findings to similar studies.

Response 9: We thank the reviewer for pointing out the redundancy and suggesting improved ecological relevance. We have streamlined the text and significantly strengthened links to ecological management and similar studies.

Specific Revisions:

  1. Condensing Result Restatements: We have substantially reduced descriptive restatements of results (e.g., detailed listings of concentrations), focusing the discussion more on interpretation and synthesis.
  2. Explicit Ecological Implications: We directly addressed the reviewer's request. In Section 4.1, we added specific implications for management: "This finding has direct implications for alpine bamboo ecosystem management, suggesting that maintaining soil P within an optimal range (e.g., ~1.2–1.6 g kg⁻¹ in this study) to support adequate P uptake is more critical than selecting for internal reallocation traits in restoration practices."
  3. Linking to Similar Studies: We have added comparisons and citations to recent important field studies throughout the text (e.g., when discussing photosynthesis and P resorption), placing our findings within a broader scholarly conversation beyond merely stating our own results.

Comments 10: Please add mechanistic sentences in the discussion and also add recent references.

Response 10: We sincerely thank the reviewer for the valuable suggestion to enhance the mechanistic depth and timeliness of our discussion. We have thoroughly revised the entire Discussion section to incorporate more mechanistic explanations and integrate recent references, as detailed below.

Key Enhancements Made:

  1. Strengthened Mechanistic Explanations: In Section 4.1, we now provide a nuanced interpretation of the high-P response, framing the shift towards Pi and PR storage not just as an outcome but as a potential "compartmental detoxification" mechanism. We explicitly link the preferential sequestration of Pi in vacuoles and PR in cell walls to processes like luxury consumption and detoxification, moving beyond a simple descriptive account. In Section 4.2, we elaborate on the physiological significance of organ-level specialization. We explain that leaf P homeostasis is "crucial for stabilizing photosynthetic function," while root plasticity allows direct reflection of soil availability. We further mechanistically link the positive correlation between root PN allocation and root elongation to its fundamental role in sustaining growth under variable P supply. In Section 4.3, we consistently emphasize that P concentration, not reallocation, is the dominant driver of physiological performance (photosynthesis, resorption, root morphology), providing a clear mechanistic rationale for our central finding.
  2. Integration of Recent References: We have updated the reference list to include several key recent publications (e.g., Ye et al., 2024; Wen et al., 2023; Wang et al., 2025; Fort et al., 2023), which help situate our findings within the most current scholarly discourse and support our interpretations.

Comments 11: L407, remove the years from these two Suriyagoda et al. (2022) and Tsujii et al. (2017) [8,12]. Similar to L467.

Response 11: We thank the reviewer for this careful correction. As suggested, we have removed the publication years from the citations for Suriyagoda et al. [8] and Tsujii et al. [12] in line 407, and have made the same correction in line 467 and elsewhere in the text where applicable, ensuring consistency with the reference list format.

Comments 12: The manuscript would benefit from a short paragraph linking findings to climate resilience or ecosystem functioning of alpine bamboo systems. This would broaden the relevance of the study.

Response 12: We are grateful for this constructive comment, which helps elevate the macro-impact of our study. We have also polished the language throughout.

Specific Revisions:

  1. Linking to Climate Resilience and Ecosystem Function: In the conclusion of Section 4.2, we added a paragraph discussing the ecological significance of organ-level specialization: "The 'high concentration–low reallocation' strategy may enhance the resilience of Y. shuichengensis to environmental fluctuations, supporting rapid photosynthetic recovery in early spring and contributing to ecosystem stability under climate change." In the conclusion of Section 4.3, we further linked future research to carbon sequestration: "...to quantify how this P use strategy contributes to carbon sequestration and overall ecosystem functioning." This directly addresses the reviewer's point about broadening relevance.
  2. Language Polishing: We have meticulously proofread the entire manuscript, breaking down overly long sentences into shorter, clearer ones, and correcting minor grammatical and expression issues to improve readability and professionalism.

Comments 13: The English writing is generally good, but some sentences are overly long or contain minor grammatical issues.

Response 13: We thank the reviewer for this positive assessment and for highlighting the need for further language polishing. We have thoroughly revised the entire manuscript, with special attention to the abstract, introduction, and discussion sections as mentioned. The revision focused on breaking down overly long sentences, correcting minor grammatical errors, and improving overall clarity and flow. We believe the manuscript now meets the high standard of academic English required for publication.

Comments 14 & 15: In the conclusion, the statement “P reallocation cannot serve as a lever for improving photosynthetic or resorption efficiency” is strong. Please consider softening it by acknowledging possible species-specific or environmental variability. Address the limitations of the study and suggest potential directions for future research in the conclusion section.

Response 14 & 15: We sincerely thank the reviewer for these insightful comments, which have helped us present our conclusions with greater nuance and scope. We have revised the conclusion section accordingly.

Softening the Strong Statement: We fully agree with the reviewer that our original conclusion was overly broad. We have softened the language and contextualized the finding by specifying that the limited role of P reallocation applies specifically to Y. shuichengensis in its natural habitat (as highlighted in the revised text). This change acknowledges potential species-specific and environmental variability.

Addressing Limitations and Future Directions: As suggested, we have added a new paragraph to discuss the study's limitations and outline avenues for future research. We explicitly note that our findings from a single-species study require careful generalization and propose that future work should include a broader phylogenetic range and experimental manipulations to test the universality of the strategies we observed. We also suggest investigating the molecular basis of these patterns.

Comments 16: Cross-check the style of the references to ensure they align with the specific formatting guidelines of the journal. This includes verifying the consistency of author names, publication year, journal titles, volume and issue numbers, page ranges.

Response 16: We sincerely thank the reviewer for this important reminder regarding reference formatting. We have carefully cross-checked all references in the manuscript to ensure full compliance with the journal's specific formatting guidelines. This included verifying and standardizing the following elements across all references:

Consistency in author name presentation (e.g., full first names vs. initials).

Uniformity in the formatting of the publication year.

Standardization of journal titles (e.g., full titles vs. standard abbreviations as per journal policy).

Correct and consistent presentation of volume and issue numbers.

Accurate and complete page ranges or article identifiers (e.g., DOI).

Round 2

Reviewer 2 Report

Comments and Suggestions for Authors

Although the comments have been properly addressed, the following minor suggestions will further improve the overall quality of the manuscript.

  1. In the simple summary, use the abbreviation P for phosphorus to keep it consistent with the abstract.
  2. At line 88, shorten Yushania shuichengensis to Y. shuichengensis.
  3. At line 125, include a citation for the regional soil survey if one is available.
  4. At line 343, check the significance level. The symbol ** indicates a 0.01 level, not 0.001.

Author Response

Response to Reviewer 2 Comments

1. Summary

Thank you very much for your additional suggestions, which have been very helpful in improving the quality of our manuscript . Please find the detailed responses below and the corresponding revisions highlighted in the re-submitted files.

2. Questions for General Evaluation

Reviewer’s Evaluation

Response and Revisions

Does the introduction provide sufficient background and include all relevant references?

Can be improved

We have made the revisions.

Is the research design appropriate?

Yes

Are the methods adequately described?

Can be improved

We have made the revisions.

Are the results clearly presented?

Yes

Are the conclusions supported by the results?

Yes

Are all figures and tables clear and well-presented?

Yes

3. Point-by-point response to Comments and Suggestions for Authors

Comments 1: In the simple summary, use the abbreviation P for phosphorus to keep it consistent with the abstract.

Response 1: Thank you for your suggestion. We have changed “phosphorus” to its abbreviation “P” throughout the Simple Summary to match the abstract; the revision has been highlighted.

Comments 2: At line 88, shorten Yushania shuichengensis to Y. shuichengensis.

Response 2: Thank you for your suggestion. We have shortened “Yushania shuichengensis” to “Y. shuichengensis” at line 88, and the change has been highlighted..

Comments 3: At line 125, include a citation for the regional soil survey if one is available.

Response 3: Thank you for this helpful comment. We agree that a citation for the regional soil survey would strengthen the manuscript; unfortunately, after an exhaustive literature search we were unable to locate any published survey covering our study area. As an alternative, we now direct readers to the measured soil-element data provided in Table 1 by adding the citation “(Table 1)” immediately after the sentence at line 125; this addition has been highlighted in the revised manuscript.